# A fiber-deprived diet disturbs the fine-scale spatial architecture of the murine colon microbiome

Alessandra Riva [1], Orest Kuzyk[1,6], Erica Forsberg [2,3], Gary Siuzdak [2], Carina Pfann[1], Craig Herbold[1], Holger Daims[1], Alexander Loy [1], Benedikt Warth [2,4] & David Berry [1,5]*

Compartmentalization of the gut microbiota is thought to be important to system function, but the extent of spatial organization in the gut ecosystem remains poorly understood. Here, we profile the murine colonic microbiota along longitudinal and lateral axes using laser capture microdissection. We found fine-scale spatial structuring of the microbiota marked by gradients in composition and diversity along the length of the colon. Privation of fiber reduces the diversity of the microbiota and disrupts longitudinal and lateral gradients in microbiota composition. Both mucus-adjacent and luminal communities are influenced by the absence of dietary fiber, with the loss of a characteristic distal colon microbiota and a reduction in the mucosa-adjacent community, concomitant with depletion of the mucus layer. These results indicate that diet has not only global but also local effects on the composition of the gut microbiota, which may affect function and resilience differently depending on location.

[1] Centre for Microbiology and Environmental Systems Science, Department of Microbiology and Ecosystem Science, Division of Microbial Ecology, University of Vienna, Althanstrasse 14, 1090 Vienna, Austria. [2] The Scripps Research Institute, Scripps Center for Metabolomics and Mass Spectrometry, 10550 North Torrey Pines Road, La Jolla, CA 92037, USA. [3] Department of Chemistry and Biochemistry, San Diego State University, 5500 Campanile Drive, San Diego, CA 92182, USA. [4] Department of Food Chemistry and Toxicology, University of Vienna, Währingerstraße 38, Vienna, Austria. [5] Joint Microbiome Facility of the Medical University of Vienna and the University of Vienna, Vienna, Austria. [6]Deceased: Orest Kuzyk. *email: berry@microbial-ecology.net

Spatial organization of microbes is ubiquitous in natural and engineered ecosystems ranging from deep-sea methane vents to thermophilic sludge granules[1,2]. Intimate and specific physical associations between diverse microbes in complex consortia can facilitate interactions important to community function, such as interspecies metabolite exchange[3] and quorum sensing[4]. Mammals have distinct microbial communities in the small intestine, caecum, and large intestine[5], which is thought to be due to local environmental variation, such as chemical and nutrient gradients as well as compartmentalized host immune activity[5–8]. The maintenance of gut health relies upon a delicate balance between diet, the commensal microbiota and the mucosa, including the epithelium and the overlying mucus layer. Intestinal goblet cells secrete a layer of mucus ~50–100 μm thick, with some reports of thicknesses reaching several hundred micrometers, that acts as a barrier between the gut lumen and the host tissue and as a habitat for microorganisms[9,10]. The mucus-associated microbiota has been suggested to be particularly important for nutrient exchange, communication with the host, development of the immune system and resistance against invading pathogens[11]. Though various factors, such as host genotype, immune status, and health state, can affect the composition of the gut microbiota, a main driver is host diet[7]. Dietary fibers are a heterogeneous class of components that are not hydrolysed by digestive enzymes of animals, and consequently are the main substrates for bacterial fermentation in the gut[12]. Dietary fiber has a major influence on gut health, and removal of dietary fiber has been shown in gnotobiotic mice to affect microbiota organization and susceptibility to enteric infection[13].

Most gut microbiota studies rely on sampling of stool, which can be obtained easily and non-invasively, but which gives no insights into spatial variation of the microbiota in the digestive tract[8]. As the extent of spatial organization and the principles that govern the gut ecosystem remain poorly understood, quantitative measurements of spatial organization at refined scales are critical to define the physical associations that underlie microbiota interactions and activities[14–16].

In order to determine the extent of fine-scale spatial structuring in the colon, we used laser capture microdissection (LCM) of colon sections to comprehensively sample small (100 x 100 x 10 μm) volumes of the complex colon microbiota and characterized these local communities with 16S rRNA gene-targeted amplicon sequencing and metagenomics. We also employed fluorescence in situ hybridization (FISH) and digital image analysis to visualize selected bacterial taxa at a single-cell level as well as global metabolomics and nanostructure imaging mass spectrometry (NIMS) to associate diet, microbes and metabolites to gain further insights into structuring of gut communities. With this approach, we observe marked variation in microbiota composition, particularly along the length of the colon. This spatial structuring is disturbed in the absence of dietary fiber and polysaccharides, leading to local changes along the colon and deterioration of the mucus layer.

## Results

### Suitability of laser micro-dissection for microbiome profiling.
We assessed the performance of low biomass LCM samples to robustly deliver accurate microbiome profiles by (i) assessing the technical reproducibility of profiles generated from DNA extracted from LCM samples, (ii) comparing profiles generated from high and low input DNA for PCR, and (iii) assessing the similarity of profiles generated from adjacent areas on the same section. We found that profiles generated from LCM samples have high technical reproducibility (Supplementary Fig. 1A), and that profiles generated from serial dilutions of the same

DNA over four orders of magnitude are highly similar (Supplementary Fig. 1B). In addition, samples collected from eight adjacent areas on the same section show high similarity (Supplementary Fig. 2D). These results indicate that LCM samples can be used to robustly and accurately determine microbiome composition. Low biomass samples are prone to contamination from PCR reagents, which we identified using a recently-developed contaminant identification approach and removed from the dataset (see Methods). This approach identified 208 ASVs as contaminants (out of 4783 ASVs), which on average comprised of 4% of reads from LCM samples.

In addition to typical reagent contaminants such as *Alpha-* and *Betaproteobacteria*[17], we noted that one of the identified contaminants was classified as *Acinetobacter*. As *Acinetobacter* has previously been reported to be a crypt-associated bacterium in the mouse gut[18] and has also been reported to be a PCR contaminant[17], so we attempted to detect it using FISH. However, we were unable to detect *Acinetobacter* even in a sample in which it had 40% relative abundance in sequencing libraries prior to contaminant removal (Supplementary Fig. 3), and we conclude that it was indeed a PCR contaminant.

### The colon microbiota is compartmentalized.
Recent studies from gnotobiotic mice have suggested that the composition of the intestinal microbiota varies depending on location, with different communities adjacent to the mucosal tissue and along the intestine[13–15]. In order to determine the extent of spatial structuring of a complex, natural colon microbiota, we comprehensively sampled the murine colon by dissecting 100 x 100 x 10 μm volumes of biomass in seven segments along the length of the colon and adjacent and distant to the mucosal tissue (hereafter termed "mucus" and "lumen" compartments) via LCM (Fig. 1a, Supplementary Fig. 4). In order to quantify the suitability of stool as a proxy for the colon microbiota, we compared the similarity of stool samples with LCM samples for each mouse. Almost all amplicon sequence variants (ASVs) found throughout the length of the colon and in both mucus and lumen compartments were also detectable in stool samples, but the level of detection was variable and stool was most similar to the lumen community of the distal colon (Fig. 1b).

While the net movement of biomass in the colon is clearly from the proximal to distal end, it is unclear whether the mucus-adjacent areas are a source or sink of gut microbes. To determine the extent to which directional processes affect microbiota composition, we applied asymmetric eigenvector map (AEM) modelling[19], an approach designed to evaluate the importance of source communities in shaping the composition of potentially downstream communities. A comparison of AEM models that tested the net direction of exchange of microbes along the colon indicated that for both mucus and lumen communities, proximal communities influence distal communities (ANOVA, $n = 163$, $p = 0.001$). However, mucus was not found to be a net source or sink of communities, suggesting extensive bi-directional transfer between mucus and lumen compartments. Consistent with AEM modeling results, variation partitioning into spatial components revealed that the longitudinal location explained 19.3% of variation in microbiota composition while mucus-lumen location explained only 1.7% (Fig. 1c).

An increase in ASV richness and Shannon diversity was observed along the length of the colon, with distal colon samples being more diverse than proximal or mid-colon (Fig. 1d). This was true for both mucus and lumen samples, but mucus samples had a lower richness compared to lumen samples from the same longitudinal position (ANOVA, $p = 0.03$).

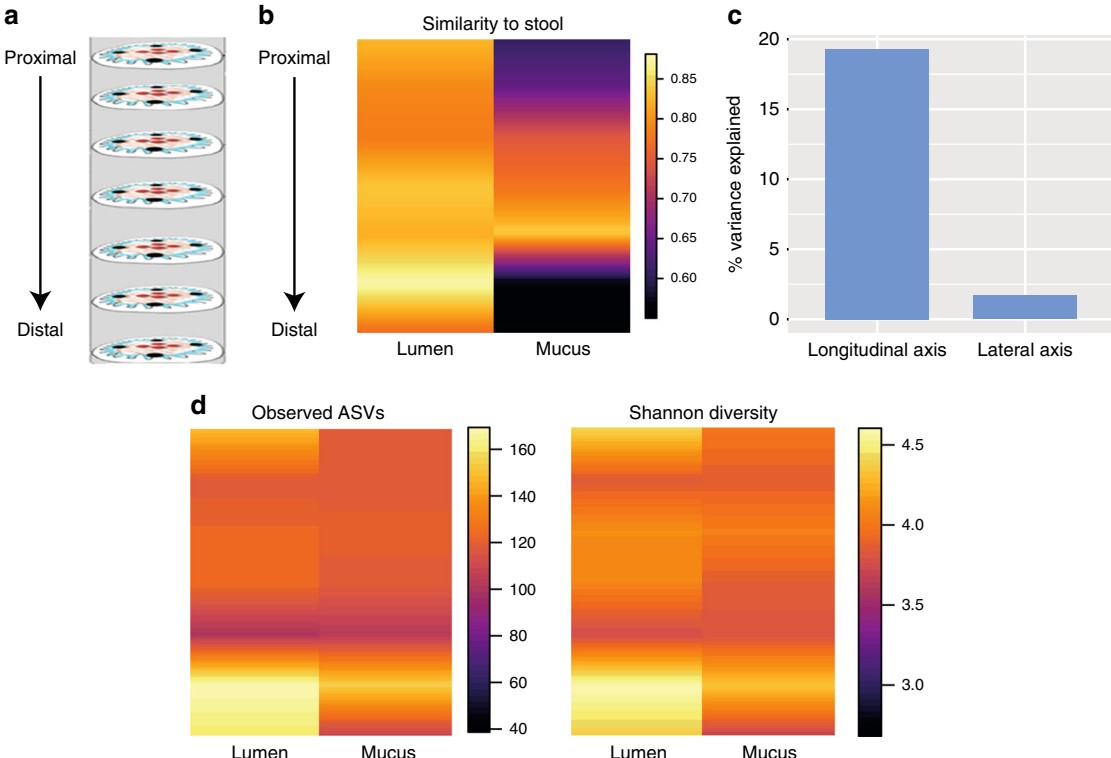

**Fig. 1** The colon microbiome is compartmentalized. **a** Sampling design. The entire colon was cryopreserved and then subsampled in seven consecutive blocks (1 cm each) along the colon. At least four areas from each section-block were sampled with the laser microdissection microscope. **b** Similarity (Bray–Curtis distance) of microbiota composition to stool in CON diet from proximal to distal colon. Interpolated average values are shown. **c** Variation partitioning analysis, which shows the percentage of variation in microbiota composition attributable to longitudinal axis (from proximal to distal colon) and lateral axis (lumen and mucus). **d** Alpha diversity heatmaps show an increase of ASV richness and Shannon diversity in the lumen compared to mucus and an increase from proximal to distal colon ($n = 211$, ANOVA and Tukey's post hoc test, mid-colon—distal colon $p < 0.001$, proximal colon—distal colon $p = 0.0027$). Interpolated average values are shown

**Deprivation of dietary fiber and polysaccharides alters the intestinal microbiome and metabolome.** Diet can have a major influence on the composition of the intestinal microbiota[13,14,16]. To address how dietary fiber and polysaccharides affect the microbiota, we compared mice given a conventional diet (CON) with mice switched for one week to either a fiber-deficient diet (FD) or a polysaccharide- and fiber-deficient diet (PFD) (Supplementary Fig. 5). The diet switch induced dramatic and rapid shifts in the composition of the microbiota, with principal coordinates analysis (PCoA) ordination showing a clear clustering of faecal pellet microbiota by diet 4 and 7 days after the diet switch (perMANOVA, $p < 0.001$; Fig. 2a). The ASV richness and Shannon diversity of the microbiota was substantially diminished in the FD diet and further reduced in the PFD diet (ANOVA, $p < 0.0001$, Fig. 2b). The composition of the microbiota was substantially altered by polysaccharide deprivation, marked most dramatically by a reduction of *Bacteroidetes* S24-7 (ANOVA, $p = 0.0002$) and by an increase in *Desulfovibrionaceae* (ANOVA, $p = 0.0002$) and *Ruminococcaceae* (ANOVA, $p = 0.0020$) (Fig. 2c).

In order to evaluate if the shifts in the microbiota were mirrored by changes in the intestinal metabolite landscape, untargeted metabolomics was performed on caecal contents collected one week after the diet shift. Metabolomics analysis revealed more than 4200 metabolic features detected with over 2100 being significantly different between the three diet groups (ANOVA, $n = 3$ for CON and PFD, $n = 2$ for FD, $p < 0.01$, min intensity 10,000) (Supplementary data 1, Supplementary Fig. 6). This large influence of diet on the metabolites present in the gut is reflected in the principal components analysis (PCA) ordination

(Fig. 3a). Notably, there was a decrease in short chain fatty acids (SCFAs) with the removal of dietary fiber, with a further decrease in some SCFAs with the removal of all dietary polysaccharides (ANOVA, Propionic acid: $p = < 0.0001$, butyric acid: $p < 0.0001$; Fig. 3b). A reduction in the SCFAs propionate and butyrate was also observed in the colon by NIMS in representative colon sections (Fig. 3c). Likewise, glutamate and N-acetyl-glutamic acid were less abundant in the contents of the mice on fiber-free diets while tryptophan and related metabolites were increased. This includes p-cresol-sulfate, a uremic toxin that was clearly more abundant in both fiber-deficient diets (ANOVA, 30.4-fold in FD [$p < 0.02$] and 12.0-fold in PFD [$p < 0.012$]). This may be of high relevance as this bacterial metabolite has been implicated in kidney and cardiovascular disease, oxidative injury and the reduction of drug action[20,21]. As expected, carbohydrate concentrations were also significantly affected by diet. For example, fructose was 1.8-fold (ANOVA, $p < 0.03$) and 3.3-fold (ANOVA, $p < 0.0014$) less abundant in FD and PFD diets, respectively, and glucose was reduced (ANOVA, 2.8-fold, $p < 0.05$) in the PFD group (Supplementary Data 1). A number of metabolic pathways were significantly changed by diet. This included glycocholate metabolism and bile acid synthesis, tryptophan and indole-3-acetate biosynthesis as well as sucrose degradation (Supplementary Fig. 6).

**Diet affects spatial structuring of the colon microbiota.** Using the LCM approach described above, we next evaluated if diet affects spatial structuring of the colon microbiota. Microbiota

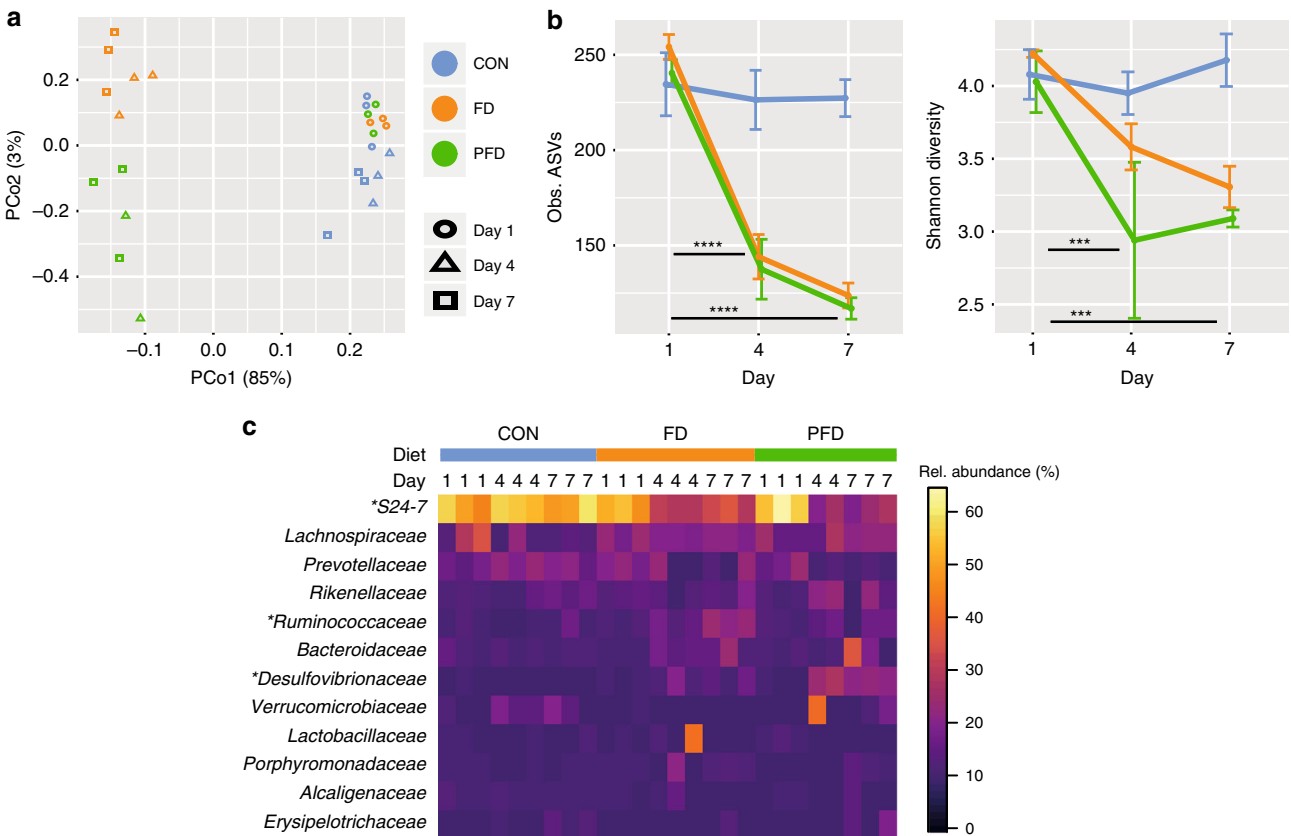

**Fig. 2** Deprivation of dietary fiber and polysaccharides alters the intestinal microbiome. **a** Principal component analysis of faecal microbiota shows a grouping of samples based by diet and day. CON = blue, FD = orange, PFD = green, Day 1 = circle, Day 4 = triangle, Day 7 = square. **b** Alpha diversity of faecal microbiota over time. At day 7 and 4 compared to day 1 FD and PFD diets had reduced alpha diversity ($n = 26$, ANOVA, PFD Obs. Species: $p < 0.0001$, PFD Shannon: $p = 0.0080$, FD Obs. Species: $p < 0.0001$, FD Shannon: $p = 0.0020$). No statistical difference was found in CON (ANOVA, Obs. species: $p = 0.6457$, Shannon: $p = 0.3136$). CON = blue, FD = orange, PFD = green. Error bars represent standard deviation of the mean. **c** Relative abundances of bacterial families in stool samples. Abundant taxa (mean relative abundance of >1%) are shown and taxa with significant changes in abundance (day 1 vs. day 7) are indicated with asterisks. (ANOVA, PFD: *Bacteroidales* S24-7, $p = 0.0009$, *Desulfovibrionaceae* $p = 0.0002$. Source data are provided as Source Data file. FD: *Bacteroidales* S24-7 $p = 0.0002$, *Ruminococcaceae* $p = 0.0020$. CON show no significance between day 1, 4, and 7 of the previously reported taxa (ANOVA: *Bacteroidales* S24-7, $p = 0.7846$, *Ruminococcaceae* $p = 1.011$, *Desulfovibrionaceae* $p = 0.3901$)

composition was found to be dependent on sampling location (both longitudinal and cross-sectional) and influenced by diet in a location-dependent manner (perMANOVA, $p < 0.001$). PCoA analysis revealed a clustering of samples by diet, with some clustering of distal and mid-colon communities in CON that was lost in other diets (Fig. 4a, Supplementary Fig. 1C and Supplementary Fig. 2). Variation partitioning of community composition revealed that diet explained 25.5% of the variation in the microbiota composition, with longitudinal location explaining 7% and mucus-lumen location explaining only 1.2% (Fig. 4b). Analogous to the stool samples, microbiota richness was decreased throughout the colon in the FD diet and even further decreased in the PFD diet. Diet affected the richness along the length of the colon in both mucus and lumen compartments (Fig. 4c, Supplementary Fig. 2), indicating that the richness of the mucus-associated community is affected by diet similarly to the lumen community. As for the CON diet, mucus samples from FD and PFD diets had lower richness than their respective lumen samples (ANOVA, $p = 0.03$). Unlike in the CON diet, ASV richness and Shannon diversity were not significantly changed along the length of the colon for either FD and PFD diets (Fig. 4c, Supplementary Fig. 2).

In the CON diet the proximal colon was enriched in *Flavobacteriaceae* (ANOVA, $p = 0.024$) and the distal colon was enriched in *Prevotellaceae* (ANOVA, $p = 0.0040$) and

*Lactobacillaceae* (ANOVA, $p = 0.024$). The lumen was enriched in *Bacteroidales* S24-7 (ANOVA, $p < 0.0001$), *Bacteroidaceae* (ANOVA, $p = 0.029$), *Prevotellaceae* (ANOVA, $p = 0.0125$), *Ruminococcaceae* (ANOVA, $p = 0.012$), *Lactobacillaceae* (ANOVA, p>0.0001) and *Erysipelotrichaceae* (ANOVA, $p = 0.016$), whereas the mucus compartment was enriched in *Lachnospiraceae* (ANOVA, $p = 0.0095$) and *Flavobacteriaceae* (ANOVA, $p = 0.012$). The pattern of enrichment of *Prevotellaceae* and *Lactobacillaceae* in the distal colon was lost in FD and PFD diets, and was accompanied by changes in the abundance of several other taxa in both longitudinal and lateral axis (Fig. 4d).

The decreased and changes in richness in microbiota caused by fiber-deficient diets was also reflected in the metabolite data that was obtained from caecal samples. When comparing the features derived from small molecules below an m/z of 200, the overall number was reduced by about 23% in the fiber-deprived diets, indicating a reduction in metabolized carbohydrate matter or microbiota-related metabolites. The mucus layer was thinner in both FD and PFD diets and fewer goblet cells were detected in the intestinal tissue (Fig. 5a). Using FISH, we found that there was an enrichment of *Lachnospiraceae* adjacent to the mucus in all diets (Fig. 5b, c). This enrichment extended into the lumen for 30 μm for CON, 20 μm for FD, and 12 μm for PFD diets (ANOVA, $p < 0.05$).

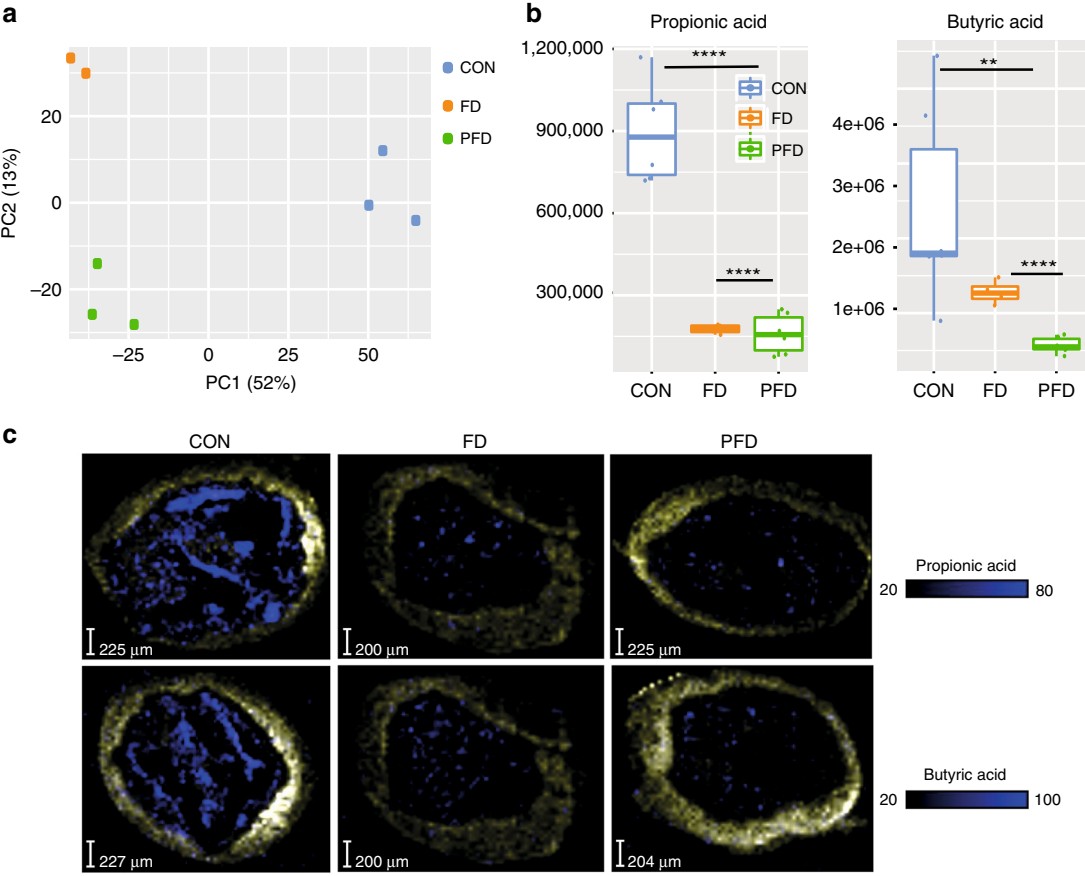

**Fig. 3** Deprivation of dietary fiber and polysaccharides alters the intestinal metabolite landscape. **a** PCA ordination illustrating that diet affects the caecal metabolite profile, as determined by global mass spectrometry-based metabolomics experiments. CON = blue, FD = orange, PFD = green. **b** Boxplots of selected metabolites in mouse caecum contents (in arbitrary peak intensity units). Boxplot: boxplot medians (center lines), interquartile ranges (box ranges), whisker ranges. (ANOVA: Propionic acid: $p = < 0.0001$, $n = 16$; butyric acid: $p < 0.0001$, $n = 16$). **c** Nanoparticle imaging mass-spectrometry of colon sections, demonstrating a decrease in propionate and butyrate in FD and PFD diets. Both propionic and butyric acid were analyzed as the K+ adduct at 113.0000 and 127.0156 m/z, respectively. Cell membrane was visualized by the mass of phosphocholine (yellow) at 184.0733 m/z. The scale for each analyte represents binned signal intensity for each 50 μm spot

**Diet-dependent spatial patchiness of the microbiota.** In land-scape ecology, habitat loss can lead to community fragmentation and spatial patchiness of communities, which is a non-homogenous distribution of species across space[22]. We there-fore tested whether deterioration of the nutrient landscape in the colon due to deprivation of dietary fiber and polysaccharides influences the patchiness of the microbiota. To determine com-munity patchiness, we applied Taylor's power law analysis, which evaluates the relative variability of all ASVs across sampling sites[23]. We found that the spatial distribution of the microbiota in the mucus is patchier than the lumen and there is a trend, though not statistically significant for FD and PFD diets to be patchier than the CON diet. (Fig. 6a). To confirm these results, we ana-lyzed selected bacterial population by FISH. We targeted *Desul-fovibrio* spp., which were enriched in PFD diet, and *Lachnospiraceae*, which were more enriched in mucus-adjacent samples and include important mucin-associated bacteria as *Eubacterium rectale* and *Clostriudim* cluster XIVa[11]. FISH revealed extensive aggregation of cells in the PFD diet (*Lach-nospiraceae*: 20% in the mucus and 28% in the lumen; *Desulfo-vibrio* spp: 2.1% in the mucus and 2.4% in the lumen), which may contribute to overall community patchiness (Fig. 6b, Supple-mentary Figs. 7 and 8).

To determine the potential functional consequences of community patchiness, we performed metagenomic analysis of distal colon samples from CON and FD diets. Consistent with the 16S rRNA gene analysis, gene richness was lower in the mucus of CON diet relative to the lumen (ANOVA, $p = 0.0018$, Fig. 6c). The difference in gene content between samples was also increased in the FD diet, and was higher in mucus than lumen, indicating increased patchiness at the gene-coding capacity (ANOVA, $p = 0.0055$ Fig. 6d). Additionally, the number of genes per KEGG functional category was also decreased (ANOVA, $p = 0.0024$, Fig. 6e), suggesting reduced functional redundancy of local communities. The overall diversity of carbohydrate-active enzymes genes was reduced in mucus-proximal regions in the CON diet as well as in the FD diet, with relative enrichment and depletion of several CAZY families (FDR < 0.1; Supplementary Fig. 9). Consistent with metabolomics data, there was a decrease in gene richness for KEGG categories involved in the production of acetate, propionate, and butyrate (Supplementary Figs. 10 11).

## Discussion

Microbial biogeography is governed in part by local environ-mental conditions. Generally, factors such as pH, oxygen, and nutrient availability are major selective forces determining the biogeography of host-associated microorganisms[24]. To determine the spatial organization of the colon microbiota, we adapted a high-spatial-resolution approach[25] to sample the microbiota

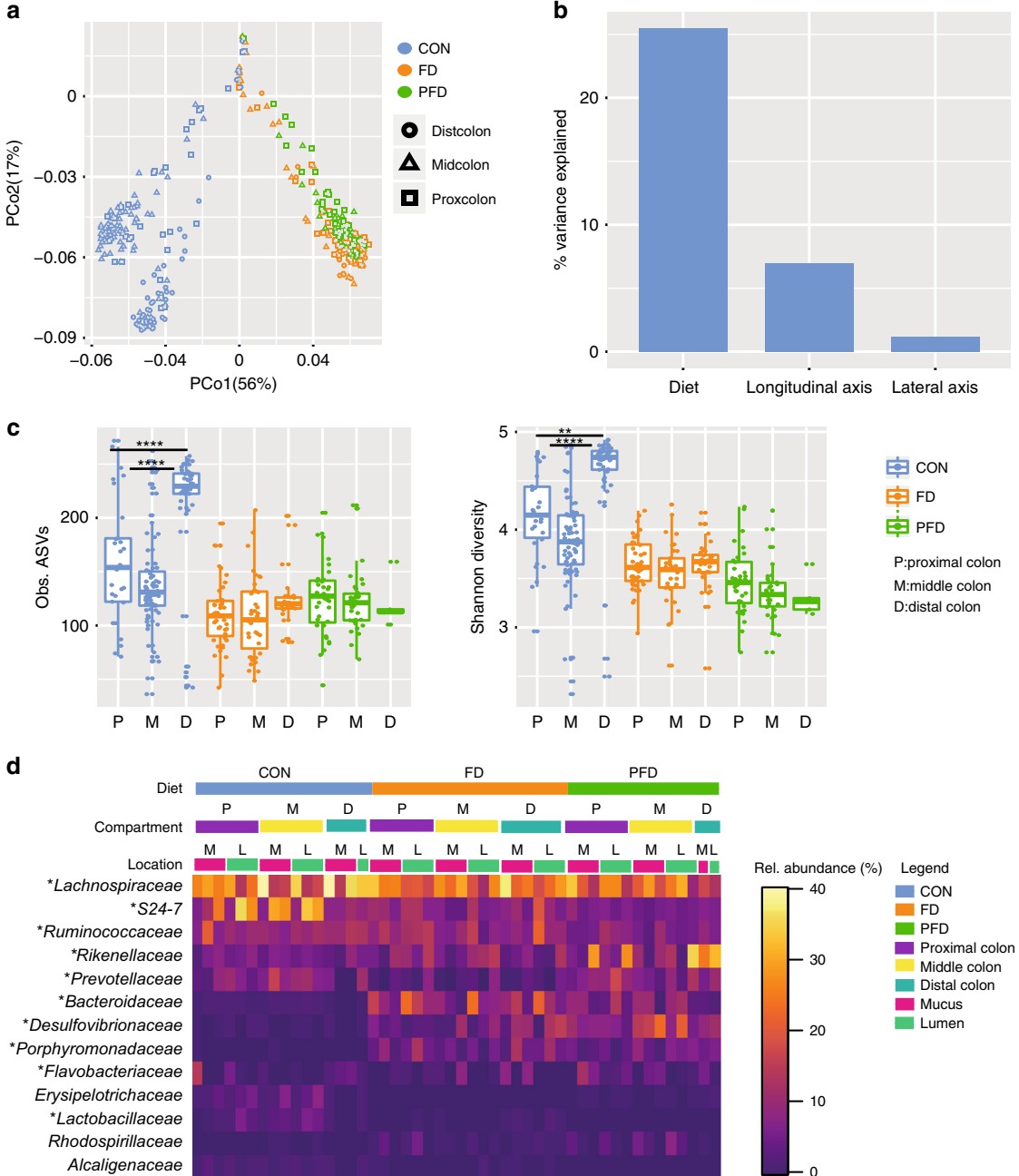

**Fig. 4** Diet affects spatial structuring of the colon microbiota. **a** PCoA of 16S rRNA gene amplicon sequence data shows grouping of samples by diet and for the CON diet location in colon (proximal, middle and distal, $p < 0.001$). CON = blue, FD = orange, PFD = green, distal colon = circle, middle colon = triangle, proximal colon = square. **b** Percentage of variance explained by diet, longitudinal axis (from proximal to distal) and lateral axis (lumen and mucus). **c** Observed species and Shannon diversity show significant differences along the length of the colon between CON vs. FD and PFD ($n = 350$, ANOVA: $p < 0.0001$), but no significant differences between PFD and FD. ASVs richness and Shannon diversity show a significant difference only in CON (ANOVA Obs. species: $p < 0.0001$, Shannon: $p = 0.0014$), whereas no significant changes were found for either FD (ANOVA, Richness: $p = 0.9687$, Shannon: $p = 0.5000$) and PFD (ANOVA, $p = 0.5829$, Shannon: $p = 0.6685$) diets. CON = blue, FD = orange, PFD = green, P = proximal colon, M = middle colon, D = distal colon. Boxplot: boxplot medians (center lines), interquartile ranges (box ranges), whisker ranges. Source data are provided as Source Data file. **d** Relative abundances of bacterial families in LCM samples. Abundant taxa (mean relative abundance of >1%) are shown and taxa with significant changes are indicated with asterisks. In FD and PFD diets *Desulfovibrionaceae* increased respect to CON (ANOVA, FD: $p = 0.005$, PFD: $p = 0.0002$) and *Lactobacillaceae* decreased (ANOVA, FD, PFD: $p = 0.0058$), in a greater extent in the distal colon. The enrichment of *Flavobacteriaceae* in the proximal colon in CON was lost only in FD diet (ANOVA, $p = 0.0047$). *Bacteroidales* S24-7 (ANOVA, FD, $p = 0.0014$), *Bacteroidaceae* (ANOVA, FD, $p < 0.0001$) *Porphyromonadaceae* (ANOVA, FD $p = 0.0031$, PFD $p = 0.023$) *Rikenellaceae* (ANOVA, FD $p = 0.0003$, PFD $p = 0.025$), *Prevotellaceae* (ANOVA, FD $p = 0.0007$, PFD $p = 0.023$) and *Ruminococcaceae* (ANOVA, FD $p = 0.0007$) increased with a prominent enrichment in the lumen. Source data are provided as Source Data file

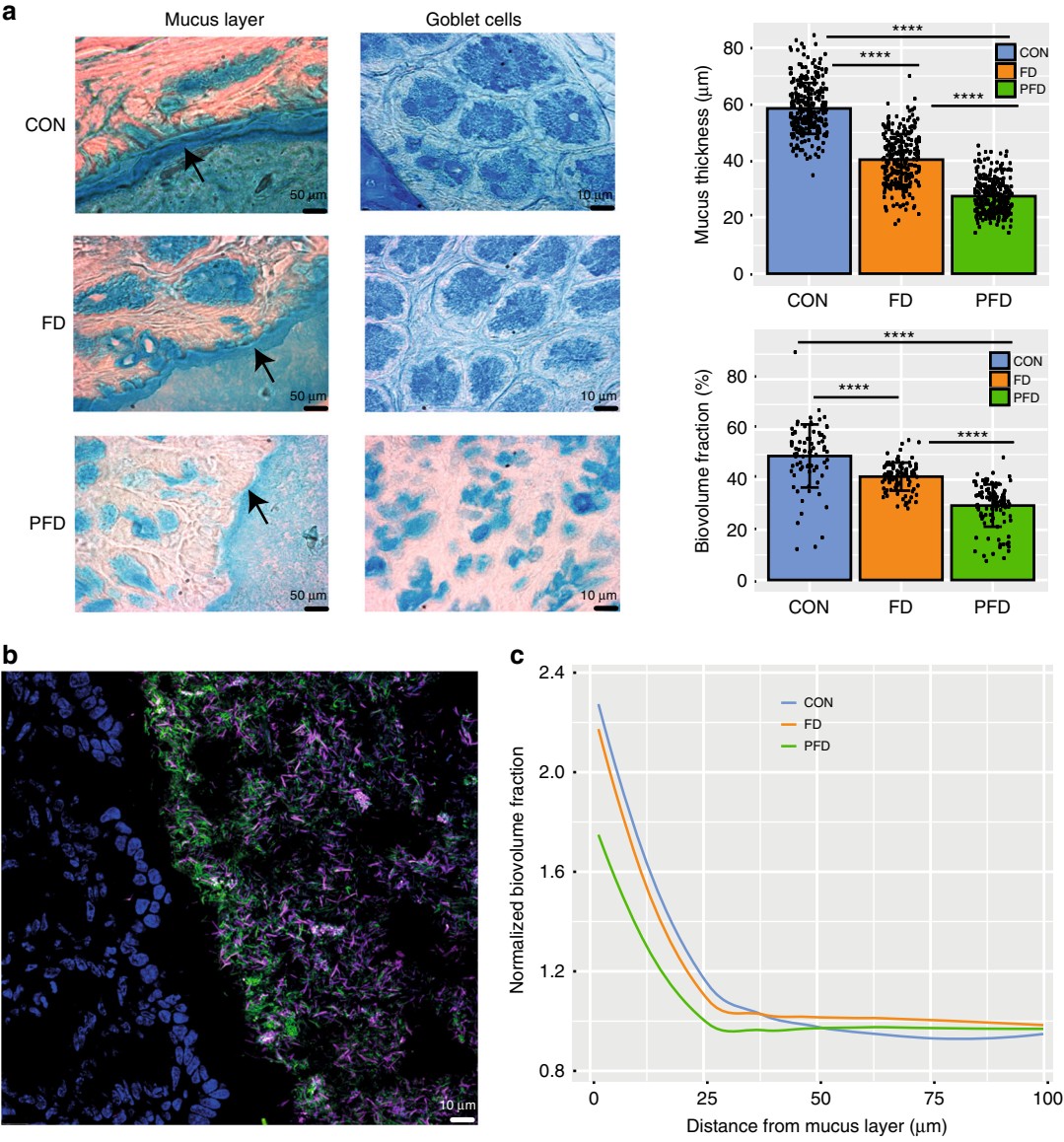

**Fig. 5** Mucus layer structure is altered by diet. **a** Representative images of the mucus layer and goblet cells stained with Alcian blue. Scale bar is 50 μm for mucus layer and 10 μm for goblet cells. Quantification of mucus layer thickness (right, top) and the biovolume fraction (right, bottom) of goblet cells in mucosa is shown. There are significant differences between all three groups [ANOVA, $p < 0.0001$ for all comparisons; $n = 841$ for mucus layer measurements (mean ± SD; CON:58.5 ± 9, FD:40.3 ± 9, PFD:27.4 ± 6.1) and $n = 289$ for goblet cells measurements (mean ± SD; CON:49.1 ± 12, FD:41.2 ± 5.5, PFD: 29.6 ± 8.4)]. CON = blue, FD = orange, PFD = green, P = proximal colon, M = middle colon, D = distal colon. Error bars represent standard deviation of the mean. Source data are provided as Source Data file. **b** Representative FISH image of a distal colon section. *Lachnospiraceae* (pink), all bacteria (green) and mucosal nuclei (blue) are shown. **c** FISH quantification of *Lachnospiraceae* biovolume fraction from the mucus into the lumen, normalized per-mouse to average biovolume fraction in the lumen ($n = 3690$ measurements)

along longitudinal and lateral axes of the murine colon. We found that the complex murine colonic microbiota exhibits fine-scale compositional gradients that are largely influenced by location along the length of the colon but also to a lesser extent the distance to mucosal tissue. In a fiber-rich diet, we found that microbial diversity increased along the length of the colon with enrichment of Proteobacteria in the proximal colon and Firmicutes in the distal colon. In line with these results, the richness of metabolites was also found to be increased. We observed that *Bacteroidetes* families were enriched in the lumen and some Firmicutes families enriched in the mucus, which is consistent with reports from biopsy samples from the human colon[22] as well as in wild-type mice[25] and in vitro models[26]. Overall, the composition of the mucus-associated microbiota was more variable

and diverse than the lumen. The mucus layer is continuously renewed by secretion from the surface goblet cells and its rapid turnover (on the order of hours) may lead to a more sparsely-colonized and dynamic environment than in the lumen, which would promote transient colonization of nearby bacteria[27,28]. Alternatively, this variability may be attributed to different stages of mucus layer colonization of distinct groups of bacteria, which has been observed in other structured environments such as dental biofilms[29]. Notably, however, we identified no taxa that were detectible only in the mucus-adjacent locations, and modeling indicated extensive exchange of bacterial populations between mucus and lumen compartments. While mucus-adjacent regions may be the source of some bacteria, these results suggest this is not a major factor in shaping lumen communities and that

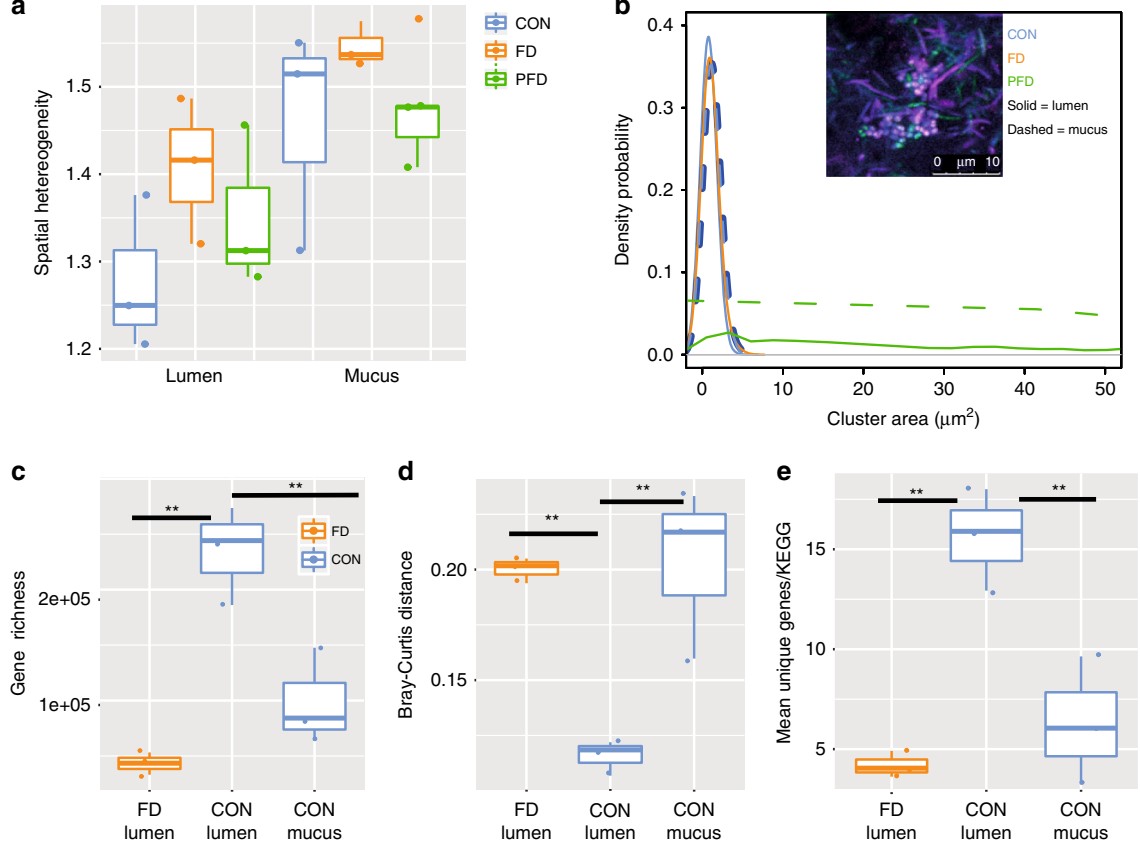

**Fig. 6** Diet-dependent spatial fragmentation of the microbiota. **a** Taylor's power analysis of the heterogeneity of microbiota composition (ANOVA, lumen vs. mucus $p = 0.004$, $n = 18$). CON = blue, FD = orange, PFD = green, P = proximal colon, M = middle colon, D = distal colon. Boxplot: boxplot medians (center lines), interquartile ranges (box ranges), whisker ranges. **b** FISH quantification of patchiness of *Lachnospiraceae*. The density probability with respect to cluster area ($\mu m^2$) in both lumen and mucus is shown. A representative FISH image shows a typical aggregation pattern in the distal colon in PFD diet *Lachnospiraceae* (pink), all Bacteria (green) and DAPI DNA staining (blue) are shown. CON = blue, FD = orange, PFD = green, solid line = lumen, dashed line = mucus. Metagenome analysis: Gene richness (ANOVA, $p = 0.0018$, $n = 9$) (**c**) Bray–Curtis distance (ANOVA, $p = 0.0055$, $n = 9$) (**d**) of samples from the same condition, and the average number of unique genes per KEGG functional category (ANOVA, $p = 0.0024$, $n = 9$) (**e**) for FD and CON are shown. CON = blue, FD = orange, PFD = green, P = proximal colon, M = middle colon, D = distal colon. Boxplot: boxplot medians (center lines), interquartile ranges (box ranges), whisker ranges. Source data are provided as Source Data file

there is extensive mixing of mucus-associated and lumen bacteria with some preferential binding or expansion of certain taxa better adapted to exploit the mucus niche[30,31]. This observation is in accordance with observations from gnotobiotic mice colonized with a 15-member microbiota that the colon does not have sharply stratified luminal and mucosal compartments but rather is better conceptualized as an incompletely mixed bioreactor[15].

Short-term fiber and polysaccharide privation dramatically and rapidly altered microbial biogeography, diversity and richness in the murine colon. Not only did loss of dietary fiber reduce bacterial diversity throughout the colon, as has been previously reported[8], but it also led to the loss of a specific distal colon community, reorganization of spatial gradients of taxa and increased spatial patchiness of communities. This was accompanied by an impairment of microbial fermentative function as determined by levels of short chain fatty acids such as propionate and butyrate. Consistent with previous studies[13,14,16], short-term elimination of dietary fiber led to mucus layer degradation and reduced mucus-producing goblet cells in the epithelial tissue. This may be due to decreased levels of butyrate, which is an important energy source for colonocytes[32], alteration in bile acid metabolism, which may alter colonic mucus secretion[33], or to increase of mucus-foraging and sulfidogenic bacteria such as *Desulfovibrio* spp., which produce $H_2S$ that attacks disulfide bonds in mucins

and makes them more accessible to degradation[34–36]. Of note, long-term privation of dietary fiber may lead to a compensatory host response and re-normalization of the mucus layer[16]. These results reinforce the notion that dietary fiber is important to colonic microbiota diversity and function but also highlight that fiber is a key driver of microbiota compartmentalization, which may have implications for local community function, microbial and host metabolism, mucus barrier function, and resilience to perturbation.

In summary, this study has established new insights into the spatial and metabolic organization and diversity of microbes in the murine colon. We showed that the complex colonic microbiota is spatially structured and that privation of dietary fiber and polysaccharides not only reduces the diversity of the gut microbiome throughout the colon, but also disrupts longitudinal and lateral gradients of microbiota composition and diversity as well as overall metabolism. This suggests a model in which dietary fiber plays a key role in shaping colonic microbiota compartmentalization and which may have important implications for host-microbial interactions.

## Methods
**Animal experiments**. Nine wild-type C57BL/6 mice maintained on a standard diet (R/M-H Ssniff, Soest, Germany) were divided in three groups ($n = 3$ per group)

and maintained for 1 week on either a polysaccharide- and fiber-rich control diet (CON) (with cellulose and starch), a fiber-deficient diet (FD) (without cellulose), or a polysaccharide- and fiber-deficient diet (PFD) (with sucrose but no cellulose or starch) (Ssniff, Soest, Germany). FD and PFD diets were isocaloric (Supplementary Fig. 5). To validate selected findings, six mice (three FD and three PFD) were included in a separate diet shift experiment (7 days). Mice were co-housed in a controlled environment (12h/12h day/night cycle) with free access to food and water. Faecal pellets were collected and snap-frozen in liquid $N_2$ on days 1, 4, and 7. Day 1 was used as baseline as the day prior to the new diet. On day 7, mice were sacrificed and the caecum contents were snap-frozen in liquid $N_2$ and the colon was cryopreserved in OCT (Thermoscientific, Massachusetts, USA) and stored at −80 °C.

The study complied all relevant ethical regulations for animal testing and research. Animal experiments were approved by the University of Veterinary Medicine, Vienna, institutional ethics committee and conducted in accordance with protocols approved by the Austrian laws (BMWF-66.006/ 0032-WF/V/3b/ 2014).

**LCM**. The colon was divided into seven 1-cm blocks and 10 μm cryosections were prepared from each block. Cryosections were mounted onto UV-treated poly-ethylene terephthalate (PET) LCM slides (Leica). LCM (Leica LCM700) was used to extract areas of ~100 × 100 μm from both mucus-adjacent areas (defined as <100 μm from epithelial tissue) as well as areas distant from the tissue (> 100μm from epithelial tissue), referred to hereafter as mucus and lumen samples, respectively. At least eight areas were sampled from each block (four for mucus and four for lumen; Supplementary Fig. 4).

**16S rRNA gene-targeted amplicon sequencing**. Nucleic acids were extracted from faecal samples using a phenol-chloroform bead-beating protocol[37]. For LCM samples, cryosections were captured in UV-treated tissue lysis buffer and extracted using the QiAmp mini DNA extraction kit (Qiagen) according to the manufacturer's instructions. PCR was performed with a two-step barcoding approach[38] using 16S rRNA gene primers targeting most bacteria (S-D-Bact-0341-b-S-17 [5′-CCTACGGGNGGCWGCAG-3′] and S-D-Bact-0785-a-A-21 [5′-GACTACHVGGGTATCTAATCC-3′].

Amplicons were quantified (Quant-iT PicoGreen dsDNA Assay, Invitrogen, California, USA), pooled, and sequenced using Illumina MiSeq at Microsynth AG (Balgach, Switzerland).

**Sequencing data analysis**. 16S rRNA gene sequence data were sorted into libraries using the eight nucleotide sample-specific barcode, quality-filtered according to the Earth Microbiome Project guidelines[38] and processed into amplicon sequence variants (ASVs) using the Divisive Amplicon Denoising Algorithm (DADA2)[39] and classified using the RDPclassifier[40] as implemented in Mothur[41]. As low biomass samples are prone to PCR reagent contaminants[17], we sequenced empty membranes, areas of membranes outside of the sample, and water, as negative controls. We followed the best practices suggested by Salter et al., including the use of UV radiation to treat the reagents, DNase treatment, the use of "blanks" controls and the same kit batch. As PCR amplification of negative control samples ($n = 10$) yielded measurable PCR products, co-elution of contaminant carrier DNA, which is used when insufficient DNA is recovered to yield a PCR product, was unnecessary. Contamination from PCR reagents and LCM membranes were identified and removed with the R package decontam (https://doi.org/10.1101/221499), which is a software specifically designed to identify contaminants in marker gene and metagenomic data from low biomass samples. The package uses a statistical test based on signatures of contamination such as prevalence and abundance of ASVs in negative control samples as compared to all other samples. For contaminant identification, the default threshold value of 0.1 was used for the prevalence-based statistical test. With this approach, we identified and removed bacterial sequences that have previously been reported to be common contamination in laboratory reagents and kits (see Results)[17].

To avoid biases related to uneven library depth, sequencing libraries were subsampled to a number of reads smaller than the smallest library (4000 reads for stool samples and 1000 reads for LCM samples). 1000 reads were sufficient to maintain a high coverage per library (mean Good's coverage: 98%). Samples with fewer than 1000 reads were removed from subsequent analysis (yielding 369 LCM samples). 16S rRNA gene sequence data has been deposited in the NCBI Short Read Archive under SRP111824.

**Metagenomic analysis**. To determine the potential function of the community, we performed metagenomic analysis of distal colon samples from CON and FD diets. DNA from LCM samples was subjected to multiple displacement amplification (REPLI-g Single cell kit; QIAgen, Hilden, Germany) to obtain sufficient amounts of DNA for sequencing. Metagenomic libraries were prepared with the TruSeq Nano DNA Library Prep Kit (Illumina) according to manufacturer's instructions. Libraries sequenced on an Illumina HiSeq 3000/4000 using 150 bp paired-end chemistry at the Biomedical Sequencing Facility (Research Center for Molecular Medicine, Medical University, Vienna). Paired-end FASTQ files were filtered using Trim Galore (software version: 0.4.1) to remove adapters and short reads (<20 bp),

and trim low-quality ends (PHRED score < 20). Assembly was done using SPAdes (software version: 3.9.0)[42] in single cell mode for MDA-amplified samples. Genes were predicted using Prokka (software version: 1.12)[43]. Protein predictions, and concatenated trimmed short reads in a separate analysis, were aligned against proteins of the mouse gut metagenome catalog[44] using RAPSearch (software version: 2.24)[45]. For annotation and taxonomy information custom scripts were used to match RAPSearch results (filtered for log10 e-value < −20, length > 50 bp, bit-score > 100, mismatch 0, identity > 99%, gap openings 0) with taxonomy and annotation information available from the mouse gut metagenome catalog. Annotation information details for KEGG ontology was retrieved using the KEGG REST API online resource. Metagenomic data has been deposited in the NCBI under SRP128037.

**Global metabolomics (LC-QTOF-MS)**. Cecal contents (50 mg) were homogenized in the presence of extraction solvent (15 μL/mg content; MeOH/$H_2O$, 4/1, v/v) and steel balls using a lab blender and then sonicated for 10 min in an ice bath. The extracts were stored at -20 °C for 30 min and centrifuged at 16,000×g (15 min, 4 °C). Supernatants were evaporated to dryness in a vacuum concentrator (Lab-conco, Kansas City, Missouri, USA) and the dried extracts reconstituted in 200 μL acetonitrile:$H_2O$ (1:1, v/v). Following another centrifugation step (16,000×g, 15 min, 4 °C), the supernatant was transferred to an autosampler vial. For each group, three biological and two technical replicates were analyzed. For the "FD group" one sample was excluded during data evaluation as inspection of the total ion chromatogram (TIC) suggested that a problem occurred during sample processing. LC-MS measurements were performed using a high-performance liquid chromatography (HPLC) system (1200 series, Agilent Technologies) coupled to a Bruker Impact II quadrupole time-of-flight (Q-TOF) mass spectrometer (Bruker Daltonics, Billerica, Massachusetts, USA). Sample extracts (4 μL) were injected onto a Luna aminopropyl column (Phenomenex, Torrance, California, USA) for HILIC analysis in ESI negative mode[46]. Data was evaluated using vendor software (Bruker Compass Data Analysis 4.3) and the XCMS Online workflow, a platform that enables online metabolomics data processing, interpretation and performs rapid metabolic pathway mapping using raw metabolomics data[47–49]. Metabolite identification was done using accurate mass comparison and matching MS2 spectra, mainly through the METLIN metabolite repository[50]. In addition, many metabolite identities were further validated by comparison with authentic reference standards. The metabolomics data was additionally evaluated using a recently-developed software algorithm for pathway analysis[51].

**NIMS with fluorinated gold nanoparticles**. To investigate the spatial distribution of small molecules in mouse colon tissue, NIMS was applied. NIMS constitutes a unique alternative to other imaging techniques such as MALDI. The fluorinated gold nanoparticles (f-AuNPs) used for this technique provide matrix-free ionization with minimal background noise. Colon sections were thaw-mounted directly onto piranha-etched p-type silicon chips. The f-AuNPs were synthetized[52] and spotted on top of the tissue slices (3 mg/mL; 1 drop per slice). Metabolites were assessed using an untargeted imaging approach in the 25–700 m/z range on a Synapt G2-Si quadrupole time of flight mass spectrometer with a MALDI source (Waters, Milford, Massachusetts, USA). Colon cryosections were prepared in 5 μm slices using a Leica 1900 cryostat (Buffalo Grove, Illinois, USA) and imaged with a 30 μm step size through oversampling using a solid-state laser with a laser diameter of 50 μm, laser energy of 175, sampling rate of 3 s, and a repetition rate of 2500 Hz. Tryptophan (m/z 205.0106) was used as a lock mass calibration standard. Metabolite identification was based on the accurate mass of the molecule in comparison with the METLIN database and the isotope pattern observed. Imaging parameters were defined using HDImaging 1.4 software (Waters, Milford, Massachusetts, USA), which was also used to process and evaluate the images. Acquisition was controlled by MassLynx 4.1.

**Alcian blue staining**. In order to visualize goblet cells and the secreted mucus layer, cryosections were stained with alcian blue (alcian blue solution 1% w/v in 3% acetic acid, pH 2.5). Samples were visualized on optical microscope (Carl Zeiss GmbH, Germany, software Axio vision 4.8, Oberkochen, Germany) and a set of 40 fields of view (×100) of the goblet cells and 35–40 fields of view (×100) of mucus were collected in order to measure the mucus layer thickness.

**FISH**. The distal colon was fixed in 4% paraformaldehyde overnight and subsequently treated with 20% sucrose overnight. The tissues were immediately frozen with isopentane (VWR, Vienna, Austria) and 10 μm cryosections were prepared. FISH was performed using a standard protocol[53]. Briefly, cryosections (two sections per slide) were quickly washed in 1x PBS and submitted to a dehydration step in an ethanol series (50–80–96%) for 3 min each. In all, 20 μl of hybridization buffer were applied with subsequent adding of 5 μM of specific probes. After 2 h of hybridization in a humidified chamber at 46 °C, the slides were washed in specific wash buffers and counter-stained with 4′, 6-diamidino-2-phenylindole (DAPI). The bacterial probes used for FISH analysis are listed in Supplementary Table 1. The NONEUB probe was used as a negative control. FISH images were taken with a confocal scanning laser microscope (Leica TCS SP8X, Mannheim, Germany). A set of 25–30 fields view for lumen and 15–20 for mucus (×63) were collected for

each mouse for each set of probes. The microscope was equipped with an Ar-laser (495 nm) for excitation of the FLUOS-dyes and two He-Ne-lasers (550 and 635 nm) for excitation of CY3 and CY5. The pinhole size was set to 1 μm and resolution used for all images ranged from 1024 × 1024–2396 × 2396 pixels and zoom factor of 1 were used for all pictures.

**Image analysis**. For each sample, 15–20 images from mucus and luminal content were taken and the biovolume fraction and spatial arrangement of cells were analyzed using the using the software daime version 2.1 program[54]. Mucus images were divided into 2 μm-thick sections from 0 to 150 μm from the mucus layer and the biovolume fraction was calculated for each section[55] (a representative example is shown in Supplementary Fig. 12). To calculate the percentage of cells in clusters, we estimated the number of cells in each cluster by measuring the size distribution of individual cells and using the mean value to convert cluster size into cell number. The thickness of the mucus layer was measured after alcian blue staining and 35–40 randomly selected fields of mucus layer and goblet cells were taken for each distal colon's mouse with a ×100 magnification. A minimum of 10 mucus thickness measurements were made per field using the program Fiji[56] and goblet cell biovolume fraction was calculated with daime[54].

**Statistical analysis**. Statistical analyses were performed using the statistical software R. The statistical significance of factors was evaluated using permutational multivariate analysis of variance (perMANOVA). Alpha diversity metrics and beta diversity were calculated using the vegan package[57]. Spatial heterogeneity in community composition was quantified using Taylor's power law, which relates the variance of the number of individuals of a species per unit area of habitat to its corresponding mean[23]. To model the distribution of the microbiota within individuals, AEM analysis was performed. The AEM framework, a new eigenfunction-based spatial filtering method, was specifically designed to model spatial structures hypothesized to be produced by directional spatial processes[19]. This spatial analytic technique was chosen as it explicitly models directional ecological processes such as the proximal-distal movement of intestinal contents. The resulting AEM vectors were forward-selected using redundancy analysis (RDA) to model species distributions.

The difference in the relative abundance between diets and compartments were performed using ANOVA for normal distributions or Kruskal-Wallis test for non-normal distributions. Post hoc tests were used for multiple comparisons between dietary groups (Tukey's and Dunn's tests after ANOVA and Kruskal-Wallis, respectively) and the paired *t*-test was used to compare the changes in relative abundance between day 1 and day 7 of dietary treatment. Variables were expressed as mean ± standard deviation (SD) and for multiple comparisons *p*-values were adjusted with the False Discovery Rate method. A p-value less than or equal to 0.05 were considered statistically significant. Statistical analysis of the metabolite features obtained from the global metabolomics measurements was performed by XCMS Online using multi-group and pairwise comparisons between either FD or PFD and the CON samples using a minimum fold change threshold > 1.5 with p < 0.01 and a signal intensity threshold of 10,000[58].

**Reporting summary**. Further information on research design is available in the Nature Research Reporting Summary linked to this article.

## Data availability
We declare that the main data supporting the findings of this study are available within the paper and its Supplementary Information. Metagenomic data has been deposited in the NCBI under "SRP128037". 16S rRNA gene sequence data has been deposited in the NCBI Short Read Archive under "SRP111824". The deposited metabolomic processing jobs including raw data files can be accessed via XCMS Public using the accession numbers 1153852 (Supplementary Fig. 6) and 1175182 (Fig. 3a). Data underlying Figs. 2c, 4c, 4d, 5a, 6c, 6d, 6e and Supplementary Fig. 2A, B are provided as Source Data files. All other data are available from the corresponding author upon reasonable requests.

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

## Acknowledgements

We would like to acknowledge the skillful support of Amelia Palermo (TSRI) during NIMS data processing. This work was financially supported by the Vienna Science and Technology Fund (WWTF, project LS12-001), the Austrian Science Fund (FWF; project grants P26127-B20, I 2320-B22, and J-3808) and the European Research Council (Starting Grant: FunKeyGut 741623).

## Author contributions

D.B, A.R., and O.K. conceived and designed the experiments. A.R. and O.K performed the experiments. A.R. and D.B. performed the data analysis and wrote the paper. B.W., E.K., and G.S performed the metabolomics analysis. C.P. and C.H. performed bioinformatic analyses. H.D. supported in image analysis. A.L. provided critical analysis and input. All authors have given approval to the final version of the paper.

## Competing interests

The authors declare no competing interests.
