## [Peer Review File · Nature Communications]

Reviewers' comments:

Reviewer #1 (Remarks to the Author):

The authors have addressed some of my questions, and I remain very interested in their study and ultimately support publication. However, there are several of my original questions that I do not think have been addressed that I think should be before publication:

- The control experiments added in Fig S4 are a great addition, but they do not address my original question: S4A shows that LCM is reproducible, while S4B shows that dilutions of a fecal sample are reproducible. But do the two approaches give the same result: that is, if you compare LCM of the lumen to analysis of the whole fecal sample, are they the same? If so, the authors should show this explicitly; if not, the authors need to comment on why and what this means for their analysis?
- I'm confused by their response to the Acinetobacter question: they say that the relative abundance was 0.7% on average in their response, but then they say in the paper "However, we were unable to detect Acinetobacter even in samples in which it was abundant in sequencing libraries (>40%, Figure S5)." What is this 40% referring to then? I maintain that if the fraction of Acinetobacter is ever high, then simply removing it will skew the relative abundances by a lot (something that was e.g. 30% will now become 50%). Can the authors report the maximum fraction of Acinetobacter that they detected in their samples, and report clearly what they did to renormalize?
- One of the other controls that I suggested, amplifying from two adjacent areas to examine this form of biological variability, does not appear to have been done. This seems like a very important control to me, as it would give a measure of short-range spatial noise.
- They should at least mention that 1 week of diet switch may not have been enough to allow for equilibration – this is substantially less than others in the field such as Sonnenburg perform.

Other suggestions:

- The use of the decontamination R package is a good addition, but for the non-expert reader it would be useful to have a brief description of what this package does.

Reviewer #2 (Remarks to the Author):

I really enjoyed reading this revised manuscript and I would like to commend the authors again for comprehensive and elegant work. All my concerns have been answered.

Reviewer #3 (Remarks to the Author):

The manuscript has been improved greatly during revision. However, a few remaining points need to be addressed prior to publication:

1. I could not find the data in the NCBI SRA at the provided accession. Sufficient data and meta-data need to be publicly accessible before the manuscript is accepted.

2. The use of “decontam” is better than simply removing all ASVs from the negative controls; however, the parameters used need to be included.

3. I still don't think that the statement in the abstract that “diet has not only global but also local effects on the gut microbiota and the metabolome” are supported. I would remove “and metabolome”, since the spatial structuring shown here is all related to community structure not function.

Reviewer #4 (Remarks to the Author):

The authors have addressed most of my comments. I nevertheless have the following remaining remarks:

1. Overall focus of the manuscript. Although the authors state in their revision letter that the aim of the study was to assess the importance of a fiber-rich diet in relation to the mucus layer (along the lines suggested by the title), the abstract and the paper remain largely unedited and the text still does not coherently follow this theme (see for example the first and second sentences in the abstract which are off-topic, i.e. these do not agree with the authors assertions in their revision letter). I would advise the authors to revise the manuscript by focusing on their major research question.

2. Experiments. I greatly appreciate the authors' efforts to include additional, confirmatory experiments in the revised version of the manuscript. In relation to the new paragraph in the results section (“suitability of laser-micro-dissection for microbiome profiling”), I would appreciate a more detailed analysis of possible contaminants other than highlighting only Acinetobacter-related sequences. Given the high abundance of this contaminant taxon (>40% of reads in sequencing libraries), a detailed assessment of contaminants is required and data from appropriate negative controls should be included in the paper as well as more details on how possible contaminant sequences were removed. The issue of contaminants in sequencing data from low-biomass samples is a well-recognized confounding factor and the authors should apply state-of-the-art methods to limit their impact and ensure that the presented data is reliable. For example, simply using empty membranes / water as negative controls is not sufficient as contaminants by themselves typically do not yield sufficient DNA for amplification (!). It is only in combination with the sample-derived DNA that contaminants become visible. Therefore, the authors should for example apply a known isolate at different cell counts to the membranes, cut out parts and assess the levels of foreign sequences. At present, given the issue with the Acinetobacter-derived sequences and the emphasis on the methodological advancement in relation to LCM (abstract), a large question mark hangs over the suitability of LCM and the general results and conclusions.

3. Statistical tests. The authors should state the names of the statistical tests along the corresponding levels of significance in the text. Also, the authors may want to revise their statistical analyses in relation to the numbers of samples analyzed (e.g. I doubt $n=2$ for FD allows for the stated results in terms of significance).

Reviewers' comments:

Reviewer #1 (Remarks to the Author):

The authors have addressed some of my questions, and I remain very interested in their study and ultimately support publication. However, there are several of my original questions that I do not think have been addressed that I think should be before publication:

- The control experiments added in Fig S4 are a great addition, but they do not address my original question: S4A shows that LCM is reproducible, while S4B shows that dilutions of a fecal sample are reproducible. But do the two approaches give the same result: that is, if you compare LCM of the lumen to analysis of the whole fecal sample, are they the same? If so, the authors should show this explicitly; if not, the authors need to comment on why and what this means for their analysis?

We thank the reviewer for their positive assessment of our work and for their comment. We have shown in Figure 1B that the LCM samples, particularly from the distal colon, are highly similar to the stool samples (calculated as Bray-Curtis dissimilarity), and have stated this in the text on lines 302-305 as follows:

“Almost all amplicon sequence variants (ASVs) found throughout the length of the colon and in both mucus and lumen compartments were also detectable in stool samples, but the level of detection was variable and stool was most similar to the lumen community of the distal colon). In addition, we have now added a principal component analysis of 16S rRNA gene amplicon sequence data that includes both LCM and stool samples (Figure S4C). This PCoA shows that the stool samples cluster with LCM samples by diet, highlighting that stool and LCM samples from the colon do have similar profiles and that diet is the main driver of community composition for all sample types.”

- I'm confused by their response to the *Acinetobacter* question: they say that the relative abundance was 0.7% on average in their response, but then they say in the paper “However, we were unable to detect *Acinetobacter* even in samples in which it was abundant in sequencing libraries (>40%, Figure S5).” What is this 40% referring to then? I maintain that if the fraction of *Acinetobacter* is ever high, then simply removing it will skew the relative abundances by a lot (something that was e.g. 30% will now become 50%). Can the authors report the maximum fraction of *Acinetobacter* that they detected in their samples, and report clearly what they did to renormalize?

After contaminants removal with the decontam package 13 ASVs classified *Acinetobacter* spp. were removed (having in total an average abundance of 0.1% across all LCM samples). To confirm that *Acinetobacter* was not present, we evaluated a sample with 40% abundance of *Acinetobacter* in sequencing libraries prior to contaminant removal using FISH and did not detect it, and therefore we conclude that it was indeed a PCR contaminant, in agreement with the findings of Salter et al, 2014.

We have modified the text as follows on lines 285-291:

“In addition to typical reagent contaminants such as *Alpha*- and *Betaproteobacteria* [22], we noted that one of the identified contaminants was classified as *Acinetobacter*. As *Acinetobacter* has previously been reported to be a crypt-associated bacterium in the mouse gut [42] and has also been reported to be a PCR contaminant [20], so we attempted to detect it using FISH. However, we were unable to detect *Acinetobacter* even in a sample in which it had 40% relative abundance in

sequencing libraries prior to contaminant removal (Figure S5), and we conclude that it was indeed a PCR contaminant.”

The relative abundances were calculated after contaminants removal performed with “decontam”, according to the best-practice recommendations of the authors of the decontam package (doi:https://doi.org/10.1101/221499). In order to test if removal of contaminants prior to calculation of relative abundances biased our results we removed the 43 samples with >10% contaminants and performed all the analyses with the subset of samples. We found that removal of these samples did not change the results. For example, the plots below with the subset of samples that had low contamination are similar to the complete sample set (compare to Figure 4A and C).

- One of the other controls that I suggested, amplifying from two adjacent areas to examine this form of biological variability, does not appear to have been done. This seems like a very important control to me, as it would give a measure of short-range spatial noise.

We have performed this control experiment, and show in Figure S7D the 16S profiles of 8 adjacent samples from cross-sections of the proximal colon in fiber and polysaccharide free diets. The plot

confirms the reproducibility of the 16S profiles generated from samples collected from a single cryosection. We have reported this result on lines 270-279:

“We assessed the performance of low biomass laser capture microdissection (LCM) samples to robustly deliver accurate microbiome profiles by (i) assessing the technical reproducibility of profiles generated from DNA extracted from LCM samples, (ii) comparing profiles generated from high and low input DNA for PCR, and (iii) assessing the similarity of profiles generated from adjacent areas on the same section. We found that profiles generated from LCM samples have high technical reproducibility (Figure S4A), and that profiles generated from serial dilutions of the same DNA over four orders of magnitude are highly similar (Figure S4B). In addition, samples collected from eight adjacent areas on the same section show high similarity (Figure S7D). These results indicate that LCM samples can be used to robustly and accurately determine microbiome composition.”

- They should at least mention that 1 week of diet switch may not have been enough to allow for equilibration – this is substantially less than others in the field such as Sonnenburg perform.

We have included discussion of about the possibility for long-term compensatory host response on lines 472-473.

Other suggestions:

- The use of the decontamination R package is a good addition, but for the non-expert reader it would be useful to have a brief description of what this package does.

We thank the reviewer for this comment. We have included more details about how the decontam package works and how we performed the analysis in the material and methods on lines 127-134: “Contamination from PCR reagents and LCM membranes were identified and removed with the R package decontam (<https://github.com/benjjneb/decontam>, doi:<https://doi.org/10.1101/221499>), which is a software specifically designed to identify contaminants in marker gene and metagenomic data. The package uses a statistical test based on signatures of contamination such as prevalence and abundance of ASVs in negative control samples as compared to all other samples. For contaminant identification, the default threshold value of 0.1 was used for the prevalence-based statistical test.”

Reviewer #2 (Remarks to the Author):

I really enjoyed reading this revised manuscript and I would like to commend the authors again for comprehensive and elegant work. All my concerns have been answered.

We thank the reviewer for their positive assessment of our work.

Reviewer #3 (Remarks to the Author):

The manuscript has been improved greatly during revision. However, a few remaining points need to be addressed prior to publication:

1. I could not find the data in the NCBI SRA at the provided accession. Sufficient data and meta-data need to be publicly accessible before the manuscript is accepted.

Our SRA data were already submitted and we have requested the NCBI to make the data public effective immediately.

2. The use of “decontam” is better than simply removing all ASVs from the negative controls; however, the parameters used need to be included.

As reported for Reviewer 1, we have included in an explanation of the method and the parameters used in the material and methods (lines 127-134):

“Contamination from PCR reagents and LCM membranes were identified and removed with the R package decontam (<https://github.com/benjneb/decontam>, doi:<https://doi.org/10.1101/221499>), which is a software specifically designed to identify contaminants in marker gene and metagenomic data. The package uses a statistical test based on signatures of contamination such as prevalence and abundance of ASVs in negative control samples as compared to all other samples. For contaminant identification, the default threshold value of 0.1 was used for the prevalence-based statistical test.”

3. I still don't think that the statement in the abstract that “diet has not only global but also local effects on the gut microbiota and the metabolome” are supported. I would remove “and metabolome”, since the spatial structuring shown here is all related to community structure not function.

“and metabolome” was removed from the title and the summary.

Reviewer #4 (Remarks to the Author):

The authors have addressed most of my comments. I nevertheless have the following remaining remarks:

1. Overall focus of the manuscript. Although the authors state in their revision letter that the aim of the study was to assess the importance of a fiber-rich diet in relation to the mucus layer (along the lines suggested by the title), the abstract and the paper remain largely unedited and the text still does not coherently follow this theme (see for example the first and second sentences in the abstract which are off-topic, i.e. these do not agree with the authors assertions in their revision letter). I would advise the authors to revise the manuscript by focusing on their major research question.

We thank the reviewer for the comment. We have attempted to shorten (as per journal requirements) and focus the summary on the main points of our work. We have also attempted to focus on our main points throughout the text.

2. Experiments. I greatly appreciate the authors' efforts to include additional, confirmatory experiments in the revised version of the manuscript. In relation to the new paragraph in the results section (“suitability of laser-micro-dissection for microbiome profiling”), I would appreciate a more detailed analysis of possible contaminants other than highlighting only Acinetobacter-related sequences. Given the high abundance of this contaminant taxon (>40% of reads in sequencing libraries), a detailed assessment of contaminants is required and data from appropriate negative controls should be included in the paper as well as more details on how possible contaminant sequences were removed. The issue of contaminants in sequencing data from low-biomass samples

is a well-recognized confounding factor and the authors should apply state-of-the-art methods to limit their impact and ensure that the presented data is reliable. For example, simply using empty membranes / water as negative controls is not sufficient as contaminants by themselves typically do not yield sufficient DNA for amplification (!). It is only in combination with the sample-derived DNA that contaminants become visible. Therefore, the authors should for example apply a known isolate at different cell counts to the membranes, cut out parts and assess the levels of foreign sequences. At present, given the issue with the *Acinetobacter*-derived sequences and the emphasis on the methodological advancement in relation to LCM (abstract), a large question mark hangs over the suitability of LCM and the general results and conclusions.

We thank the reviewer for this comment. We ask the reviewer to refer to our answer to Reviewer 1 regarding contaminant removal and *Acinetobacter* analysis. The contaminants were largely alpha- and betaproteobacteria that have previously been reported as reagent contaminants, and we have added these details and an appropriate reference in the Results. Regarding the use of DNA to co-elute contaminants, this was not necessary with our PCR conditions. While working with low biomass samples is a (largely underappreciated) challenge, we have been extremely careful and performed many control experiments and analyses, and we feel that these have supported that our technical and analytical approaches deliver accurate and robust results.

3. Statistical tests. The authors should state the names of the statistical tests along the corresponding levels of significance in the text. Also, the authors may want to revise their statistical analyses in relation to the numbers of samples analyzed (e.g. I doubt $n=2$ for FD allows for the stated results in terms of significance).

We thank the reviewer for the comment. We have added the statistical test used along the corresponding p-value in the text and in the legend. Unfortunately, one of the samples failed in metabolomics and no backup samples were available for re-analysis (hence $n=2$ for FD). This likely results in reduced power to detect differences, but this was also not a main focus of our study.

REVIEWERS' COMMENTS:

Reviewer #1 (Remarks to the Author):

Responses are great, I look forward to seeing the manuscript published.

Reviewer #4 (Remarks to the Author):

The authors have addressed my remaining comments through their latest revision of the manuscript. While I do appreciate their point of view, the authors do not provide a coherent argumentation for why their PCR conditions do not require co-elution of known DNA in order to comprehensively identify contaminant sequences. The authors may want to include such argumentation in the final version of the manuscript.

Signed: Paul Wilmes

Reviewer #4 (Remarks to the Author):

The authors have addressed my remaining comments through their latest revision of the manuscript. While I do appreciate their point of view, the authors do not provide a coherent argumentation for why their PCR conditions do not require co-elution of known DNA in order to comprehensively identify contaminant sequences. The authors may want to include such argumentation in the final version of the manuscript.

We thank the reviewer for this comment. The use of “blanks” as negative controls is a common practice and the use of co-elution of known DNA is not necessary as the negative control samples yield a range of contaminating bacterial species, which were measurable as PCR product, taking also into account that a standard PCR composed by 30-35 cycles is able to amplify detectable material also in the blanks/negative controls.

In addition, the use of the R package decontam made possible to better identify and remove these contaminants. We have included this detail in the methods (line 370-384).